# Optimization of an Alternative Culture Medium for Phycocyanin Production from *Arthrospira platensis* under Laboratory Conditions

**DOI:** 10.3390/microorganisms12020363

**Published:** 2024-02-10

**Authors:** Daniel Alberto Freire Balseca, Kimberly Susana Castro Reyes, María Elena Maldonado Rodríguez

**Affiliations:** Life Sciences, Biotechnology, Giron, Quito Campus, Universidad Politécnica Salesiana, Quito 170517, Ecuador; dfreireb@ups.edu.ec (D.A.F.B.); kcastror1@est.ups.edu.ec (K.S.C.R.)

**Keywords:** spirulina, phycocyanin, response surface methodology, central composite design, pigment, urea, potassium nitrate

## Abstract

*Arthrospira platensis*, known as spirulina, is a cyanobacterium with multiple nutritional benefits, as it contains substantial amounts of proteins, fatty acids, and pigments. However, the production of this microalga has faced significant challenges, primarily related to the cost and composition of the required culture medium for its optimal growth. This study focused on optimizing two nitrogen sources (urea and potassium nitrate) to maximize the growth of *A. platensis* and the production of phycocyanin, a photosynthetic pigment of significant commercial value. Optimization was performed using the response surface methodology (RSM) with a central composite design (CCD). Analysis of variance (ANOVA) was employed to validate the model, which revealed that the different concentrations of urea were statistically significant (*p* < 0.05) for biomass and phycocyanin production. However, potassium nitrate (KNO_3_) showed no significant influence (*p* > 0.05) on the response variables. The RSM analysis indicated that the optimal concentrations of KNO_3_ and urea to maximize the response variables were 3.5 g L^−1^ and 0.098 g L^−1^, respectively. This study offers valuable perspectives for the efficient production of *A. platensis* while reducing production costs for its cultivation on a larger scale.

## 1. Introduction

Spirulina is a blue-green microalga scientifically known as *Arthrospira platensis*. This aquatic species belongs to cyanobacteria. They are capable of photosynthesis and form multicellular filaments called trichomes, which are not branched and have a spiral shape. The production of *A. platensis* is highly significant in Asian countries such as India, Japan, and China [1]. This organism is of great nutritional significance due to its high protein content, ranging from approximately 60% to 70%. Additionally, it contains polysaccharides in the range of 15% to 25%, polyunsaturated fatty acids representing between 3% and 9% of dry weight, and various pigments, including chlorophyll, phycocyanin, and carotenoids [2]. In the pharmaceutical industry, spirulina plays a significant therapeutic role; human clinical trials and animal studies show that this species can be used in the treatment of cardiovascular diseases, high cholesterol, elevated blood sugar levels, obesity, inflammatory diseases, etc. [1]. Additionally, this alga contributes to strengthening the immune system and is associated with slowing the progression of neurodegenerative diseases such as Parkinson’s, Alzheimer’s, and multiple sclerosis [3,4].

Several factors influence the biomass production of *Arthrospira platensis*, such as temperature, aeration rate, CO_2_, concentration, nitrogen sources, and phosphate [5]. Nitrogen plays a crucial role in the growth and biochemical composition of *A. platensis* [6] microalgae because it is an essential component of amino acids and is therefore required for synthesizing proteins and pigments. Among these pigments are chlorophyll and phycocyanin, which are responsible for the photosynthetic activity of *A. platensis* and, consequently, biomass production [7]. In addition, nitrogen is involved in various metabolic processes, such as synthesizing nucleic acids, some lipids, and amino acids. These processes are crucial for cell division, energy storage, and growth [8].

The type and amount of the nitrogen source in the formulation of the culture media influence growth and metabolite production in *A. platensis.* The use of sodium and potassium nitrates as nitrogen sources in the production of commercial culture media has shown excellent results in microalgal biomass growth; however, previous studies evidenced benefits when using alternative and low-cost sources of nitrogen such as ammonium nitrate, urea, ammonium sulfate, and ammonium chloride [9,10,11]. In the present trial, urea has been used as an alternative nitrogen source to reduce costs in culture media preparation. This compound has demonstrated satisfactory results in cell growth, as it is metabolized more rapidly and with minimal energy expenditure by the cells [12,13]. However, a toxic effect has been evidenced when this compound is used in high concentrations, causing a decrease in the growth of cyanobacteria and, consequently, cell death [9,14].

RSM is an experimental and analytical strategy whose main objective is to solve problems by providing the optimal operating conditions for a process. The results obtained from this strategy are referred to as optimal values [15]. To optimize variables, it is necessary to implement an experimental design, which is a statistical tool that simultaneously examines factors and their different levels through a reduced number of experiments [16]. The central composite design (CCD) has been commonly used for optimizing culture media and adjusting response surfaces. CCD allows for determining the appropriate concentration of nutrients and other factors that influence the growth and metabolite production of the species under study [17].

This research aims to optimize the composition of an alternative culture medium to increase the biomass of *Arthrospira platensis* and, consequently, the production of phycocyanin, employing experimental design methods focused on the optimization of two significant factors, i.e., the two selected nitrogen sources, potassium nitrate (KNO_3_) and urea, using the response surface methodology in a central composite design to obtain the optimal composition of the alternative culture medium.

## 2. Materials and Methods

### 2.1. Biological Material and Inoculum Preparation

To initiate the experimental study, a culture of the microalga *Arthrospira platensis* from the Life Sciences Laboratories of the Universidad Politécnica Salesiana was used. To scale up the spirulina biomass before the experimental analysis, Zarrouk culture medium was employed, using 1 L Erlenmeyer flasks, a 3 W air pump with a 12:12 light–dark photoperiod, and a light intensity of 1000 µmolPhotonPAR/m²/s, for which white light lamps were used.

### 2.2. Experimental Design

To optimize the artisanal culture medium, the response surface methodology was employed, using the central composite design (CCD) technique. For this purpose, two controllable factors were defined: the initial concentrations of KNO_3_ (g L^−1^) and urea (g L^−1^).

The selected factors were analyzed at 5 levels, encoded with symbols and numbers, as shown in Table 1.

The statistical software Design-Expert version 22.0.8 (Stat Ease, Inc., Minneapolis, MN, USA) and Minitab 21.4.0 were used to generate the optimization treatments from a CCD; each treatment was conducted in triplicate. In addition, 2 central point treatments (0,0) were performed, and the alpha (α) value recorded for the axial points was 1.4142. A total of 27 experimental units were prepared, where a concentration of 0.24 gL^−1^ of spirulina biomass was inoculated for each experimental unit.

Each experimental unit was prepared in an artisanal photobioreactor using an Erlenmeyer flask containing 2 L of culture medium according to the model generated by the CCD. The analysis was carried out over 24 days under the following conditions: an average temperature of 21 °C, a 12:12 photoperiod, continuous aeration, and a light intensity of 1000 µmolPhotonPAR/m²/s. Measurements of biomass growth and phycocyanin were taken five times during the experimental period (days 0, 6, 12, 18, 24).

### 2.3. Biomass Quantification

The dry weight technique was used to quantify the biomass concentration in each experimental unit. A disassembled Kitasato, a Buchner funnel with a 0.8 µm Millipore membrane, and a Mettler Toledo HB43-S halogen moisture detector were used. Equation (1) was employed to determine the dry weight:(1)$$Biomass\;Concentration\left({gL}^{-1}\right)=\frac{mass\;of\;the\;membrane\;with\;biomass\left(g\right)-membrane\;mass\;(g)}{culture\;medium\;volume\;(L)}$$

The results obtained from this procedure were used in subsequent calculations to evaluate the growth kinetics of *Arthrospira platensis* in each treatment.

### 2.4. Quantification of Phycocyanin Concentration

To determine the phycocyanin concentration value for each treatment, three steps were performed: first, the extraction of the pigment; then, the measurement of its absorbance using spectrophotometry; and finally, the calculation of the percentage of phycocyanin.

In the extraction process, a vacuum filtration technique was employed using a disassembled Kitasato flask to obtain a quantity of wet biomass ranging from 40 to 50 mg.

Next, 10 mL of 0.2 M phosphate buffer and the pre-weighed wet biomass sample were placed in a test tube and agitated in a vortex for 5 min. After agitation, the sample was frozen at −4 °C for 24 h. After this time, the sample was sonicated for 40 min at 60 Hz. Then, the sample was centrifuged at 5000 rpm for 10 min. Finally, the supernatant containing phycocyanin was extracted and quantified using spectrophotometry. Figure 1 graphically illustrates the steps for phycocyanin extraction.

To measure the concentration of phycocyanin in the supernatant obtained from the extraction process, a JASCO V-730 spectrophotometer (Spectra Manager^TM^, Easton, EE.UU.) was used at two wavelengths: 615 nm and 652 nm. The concentration of phycocyanin was determined using Equation (2) described by Bennett and Bogorad [18].
(2)$$\left[PC\right]=\frac{A_{615}-{0.474(A}_{652})}{5.34}$$
where [*PC*] represents the concentration of phycocyanin, whose unit of measurement resulting from the equation is mg mL^−1^; *A* is the absorbance obtained from different wavelengths.

### 2.5. Quantification of the Concentration of Nitrates and Ammonia Salts

Measurements of nitrates and ammonia salts were performed using the Hach-DR900 multiparametric portable colorimeter. The residual fluid obtained from the filtration in the *A. platensis* biomass extraction process was used to perform the corresponding quantifications. The concentration (g L^−1^) of nitrates was measured using method 8039 or the cadmium reduction method [19], and the concentration (g L^−1^) of ammonia salts was measured using method 8155 or the salicylate method [20] from the Hach-DR900 procedures guide.

### 2.6. Statistical Analysis

After data collection, R software version 4.3.1 was employed to generate the growth curves of *A. platensis* over the 24 days of experimentation and comparative bar graphs to assess the relationship between ammonia and nitrate concentrations against the response variables. Additionally, Design-Expert version 22.0.8 and Minitab 21.4.0 software were used to analyze the optimization of the controllable factors (urea and KNO_3_) using the response surface methodology to achieve the best condition of the dependent variables (biomass and phycocyanin). It was determined whether the significance values p and F were satisfactory for the chosen mathematical model. The quality of the mathematical model was verified by considering the coefficient of determination (R^2^) and the adjusted coefficient of determination (R^2^_adj_).

## 3. Results and Discussion

This study aimed to optimize the concentration of urea and KNO_3_ to increase the biomass production of *Arthrospira platensis* and the percentage of phycocyanin using an alternative culture medium through the implementation of a central composite design (CCD) that includes nine treatments in triplicate.

According to Kasina et al. [21], CCD is commonly used in optimization criteria due to its ease of use and the ability to estimate all parameters in a second-order model. In the execution of the response surface design, the rotational design property was chosen. In optimization processes, a rotational design is crucial, because it allows for more information to be extracted regarding the dependent variables and reduces uncertainty in predicting responses. Additionally, the concentration of nitrates and ammonia salts in each treatment was analyzed. Table 2 displays the combinations of urea and KNO_3_, along with their respective results after 24 days of cultivation.

### 3.1. Effects of KNO_3_ and Urea on the Biomass Formation and Phycocyanin Production of A. platensis

In this study, the response surface methodology using a central composite design was employed to assess the effects of different concentrations of urea and KNO_3_ on the behavior of *Arthrospira platensis* and phycocyanin production.

It was confirmed that treatment 1 provided the highest biomass concentration, with an average value of 0.91 g L^−1^, followed by treatment 8, with an average value of 0.85 g L^−1^. In contrast, no growth was recorded in treatments 4, 5, and 9 (Figure 2). Treatments 1 and 8 consisted of 4.5 g L^−1^ of KNO_3_, 0.15 g L^−1^ of urea and 0.010 g L^−1^ of urea and 2.5 g L^−1^ of KNO_3_, respectively (Table 2). Studies conducted by Baky et al. [6] and Gómez et al. [22] indicated that using 2.5 g L^−1^ of nitrates resulted in higher biomass yield, which can be attributed to the increased availability of nitrogen in the medium. Additionally, in the study conducted by Castro et al. [23], it was mentioned that the excess nitrogen in the culture media is stored as organic compounds, such as protein pigments. This evidence was corroborated by the increase in the production of the phycocyanin protein in treatment 1, reaching the highest percentage of 15.87% (Figure 3b).

Nitrogen is an essential element for the cultivation of *A. platensis* due to its significant influence on the development of this microalga, and its effect depends on the quantity, availability, and source type [24]. Nitrates are the standard nitrogen source, and ammonium has also been used as a nitrogen source because it is a more cost-effective alternative [25]. The death of the inoculated biomass in treatments 4, 5, and 9 may result from the high concentrations of urea (0.23 g L^−1^ and 0.31 g L^−1^), generating high levels of ammonia within the medium and consequently exerting an inhibitory effect on the growth of *A. platensis* [10]. Various concentration values of urea that inhibit the growth of *A. platensis* have been documented, including the one reported by Mateucci [26], where the inhibitory concentration of urea is 0.88 g L^−1^, and in the study conducted by Madkour et al. [27], an inhibitory concentration of 0.3 g L^−1^ was observed. Torrea et al. [28] and Vieira et al. [29] reported inhibitory concentrations of 0.3 g L^−1^ and 0.5 g L^−1^, respectively. Figure 2 displays the biomass growth curves (g L^−1^) of *A. platensis* during the study period. It is observed that in the first 6 days, the inoculated biomass (0.24 g L^−1^) in treatments 4, 5, and 9 dies due to the presence of high concentrations of ammonia in the medium, where a concentration of 0.0422 g L^−1^, 0.0268 g L^−1^, and 0.0307 g L^−1^, respectively, was reported; according to Carvalho et al. [11], concentrations ≥ 1.6 mM (0.0272 g L^−1^) cause toxicity in the culture medium and, consequently, the death of the microorganism.

The quality of *A. platensis* depends on the nutrients present in the culture medium, with nitrogen being essential for its growth, as it plays a crucial role in the production of amino acids, which are fundamental components of proteins [1]. *Arthrospira platensis* cannot fix N_2_; to utilize nitrogen sources such as NO_2_ and NH^+^ ions, this cyanobacterium requires specific membrane transporters, such as NRTs (nitrate transporters). Once nitrate has entered the cells, it is reduced to nitrite by the enzyme nitrate reductase (Nar) and subsequently converted to ammonia through the nitrite reductase enzyme (Nir). Finally, ammonia is incorporated into protein structures [13].

Ureolysis is the enzymatic hydrolysis of urea by the urease enzyme present in *A. platensis*. The products obtained from this enzymatic process are ammonia and carbon dioxide. This ammonia is easily assimilated by microalgae and cyanobacteria [30].

However, the presence of high concentrations of ammonia reversibly inhibit the assimilation of alternative nitrogen sources, such as nitrates. This occurs due to the inactivation of specific nitrate transporters [31]. The ease with which ammonia penetrates *A. platensis* cells leads to inhibitory effects, because the cells cannot control ammonia absorption. As a result, intracellular accumulation of this compound occurs, ultimately leading to cell intoxication and death [32].

Figure 3a,b shows that in treatments with a high initial urea concentration (treatments 4, 5, 9), ammonia levels are proportional. The highest biomass growth and phycocyanin production were recorded in treatment 1 on day 24 (Figure 3a). Biomass growth remained consistent throughout the experimental period (Figure 2).

Mulokozi et al. [32] state that spirulina cannot grow above a urea concentration of 0.18 g L^−1^, as a strong ammonia odor is detected. Under alkaline conditions, urea decomposes into ammonia, which, in high concentrations, acts as a toxin that hinders photosynthesis and the growth of this species. This information was corroborated by the results obtained in treatments 4, 5, and 9, where no biomass growth was observed.

Phycocyanin, a water-soluble pigment, is closely related to biomass concentration, and several factors such as nutrient availability, pH, salinity, light irradiation, and agitation speed affect the growth and accumulation of this pigment in microalgae [33]. A study conducted by Castro et al. [23], which involved the optimization of a modified Zarrouk culture medium with varying concentrations of sodium nitrate and sodium bicarbonate, demonstrated that using 2.5 g L^−1^ of nitrates results in a higher concentration of phycocyanin, reaching 17.9%. This indicates that the obtained results do not significantly differ from this value. As mentioned earlier, treatments 4, 5, and 9 did not exhibit biomass production, and consequently, no phycocyanin was produced (Figure 3b). According to Mateucci [26], this phenomenon can be attributed to the presence of ammonia in concentrations exceeding 2 mM (0.034 g/L) in the culture medium, leading to toxicity. Furthermore, it has been shown that high concentrations of ammonia (NH_3_) inhibit the photosynthetic system of cyanobacteria and therefore have a detrimental impact on biomass production, protein production, and, consequently, phycocyanin production [25].

### 3.2. Model Adjustment and Analysis of Variance (ANOVA)

Based on the estimated parameters and using the RSM with a central composite design, the adjusted regression Equations (3) and (4) for biomass and phycocyanin production were obtained.
(3)$$Biomass\left({gL}^{-1}\right)=+0.590909-0.274010\;Urea+0.108425\;{KNO}_{3}-0.082500\;Urea*{KNO}_{3}-0.169773\;{Urea}^{2}$$
(4)$$Phycocyanin\left(\%\right)=+9.84268-4.53263\;Urea+1.2995\;{KNO}_{3}-0.676125\;Urea*{KNO}_{3}-2.87358\;{Urea}^{2}$$

The coefficients with a positive sign in the adjusted model indicate that the variable can improve the response. On the contrary, a negative sign in the regression equation implies the variable’s capacity to reduce the response [34]. Equations (3) and (4) indicated that an increase in urea concentration negatively affects the growth of microalgal biomass. This negative effect is directly proportional to phycocyanin production.

Regarding the presence of KNO_3_, the quadratic model reflects that as the concentration of this component increases, the response variables exhibit positive behavior, meaning that an increase in the biomass of *Arthrospira platensis* is predicted, and consequently, an increase in the production of phycocyanin occurs. According to Ruliaty et al. [35], KNO_3_ is used simultaneously with urea as a nitrogen source because it keeps the microorganism stable, thereby increasing cell concentration and the productivity of *A. platensis*.

In the study, an analysis of variance (ANOVA) was conducted as part of the model adjustment design to identify statistically significant terms. ANOVA analysis is a statistically important tool, as it determines whether substantial differences exist between the independent variables and the response variables [36]. Table 3 displays the results of the ANOVA analysis for the two response variables studied.

F-distribution is a statistical analysis used to assess differences among the studied groups and test the null hypothesis (H_0_), which suggests that the variances or means of two or more groups are equal [37]. Regarding biomass production, the analysis of variance yielded an F-value of 11.34, corresponding to the adjustment model, indicating that the model is statistically significant. When calculating the Fc_rit_ value (0.05; 4; 13), a value of 3.18 was obtained, which is lower than the model’s F-value. As a result, H_0_ (the model is not statistically significant) was rejected, ensuring that the results obtained are statistically significant with 95% confidence in the effort to maximize the growth rate of *Arthrospira platensis*.

In this study, the ANOVA analysis provided a *p*-value of 0.0005, significantly lower than the α value (0.05). Therefore, H_0_ is rejected, demonstrating that the terms included in the model are significant. However, it is worth noting that the term (KNO_3_ * KNO_3_) did not show statistical significance and was therefore not considered in the adjusted model. Similarly, the term corresponding to the interaction between factors (Urea* KNO_3_) did not show significance in the model, indicating that the concentration of KNO_3_ does not have a significant impact on biomass growth in this statistical analysis.

When comparing the Fc_rit_ value (0.05; 4; 13) = 3.18 with the model’s F-value of 10.78, it is concluded that the model for phycocyanin production is significant. In addition, the model had a p-value lower than the confidence significance value α (0.0006 < α). The term “Urea” was significant in the model, influencing the response variable.

In contrast, the concentration of KNO_3_ did not show significant effects on phycocyanin production (*p* > α), so it was removed from the quadratic regression model. In addition, the model’s validity was verified through a lack of adjustment analysis, which indicated that the model accurately fitted the data, with a *p*-value of 0.7146 greater than 0.05.

The significance p and F values indicated that the models could provide reliable experimental data.

### 3.3. Adjustment Quality

To assess the quality of the adjustment of specific models, the coefficients of determination (R^2^) and adjusted determination (R^2^_aj_) were calculated. These values allow for measuring the accuracy with which response values are predicted and the ability of the regression models to describe the data [15]. Table 3 displays the coefficients of determination (R^2^) for each adjusted quadratic model, the biomass growth model and the phycocyanin production model, with results of 0.7907 and 0.7822, respectively. In addition, the adjusted coefficient of determination (R^2^_aj_) was calculated for the mentioned quadratic models. The coefficient value for biomass growth was 0.721, and for phycocyanin production, it was 0.7096. According to Pulido et al. [15], the values of R^2^ and R^2^_aj_ indicate the quality of the model fit, as they explain the data variation. A value exceeding 0.7 in both coefficients demonstrates the model has a satisfactory fit. The values obtained for each coefficient validated the reliability of the models in estimating the optimal concentrations of urea and KNO_3_ required to maximize biomass growth and phycocyanin production.

Additionally, normal probability plots of residuals were examined to determine whether the model exhibited a satisfactory adjustment. In Figure 4, plots (a) and (b) display the normal probability plots of residuals for the described models.

The normal probability plot assesses whether a dataset follows a normal distribution. This plot is employed to identify significant deviations, outliers, and kurtosis [38]. It was observed that the residuals are uniformly distributed around the normal line (diagonal line), indicating statistical kurtosis, also known as 0 value, which means there is no systematic bias in any direction. Statistical kurtosis refers to the extent to which the data are concentrated around the mean of the distribution [39]. This graph suggests that when used accurately, the model reflects the data’s variability; therefore, there are no systematic errors in the estimation.

### 3.4. Response Surfaces and Contour Plots for Optimizing Response Variables: Arthrospira platensis Biomass and Phycocyanin

To visualize the main effects and interactions of the variables used in the growth of the *Arthrospira platensis* biomass, a response surface plot and a contour plot (Figure 5a,b) were employed, representing the concentration of urea and KNO_3_ in relation to the response variable. It was observed that the biomass gradually decreases as the urea concentration increases; a high urea concentration can lead to biomass death in the culture medium. On the other hand, the effect of KNO_3_ was reciprocal to the gradual decrease in biomass in response to the urea factor. Additionally, it was evident that the KNO_3_ factor positively affects biomass production. However, statistically, this factor is not significant.

Phycocyanin exhibited a directly proportional relationship with biomass growth. According to Castro et al. [23], biomass productivity is directly related to phycocyanin concentration due to the storage of nitrogen in the form of organic compounds, such as proteins (phycocyanin). Figure 5c,d displays the response surface plot and contour curves for phycocyanin production, illustrating the effects of urea and KNO_3_ concentrations. The graphs show that as the urea concentration increases, phycocyanin production is negatively affected. This indicates that low concentrations of urea have a positive effect on phycocyanin production. Regarding the KNO_3_, the graphs reveal that its influence on phycocyanin production is not statistically significant, as the curves follow the same trend with different urea concentrations.

Based on the findings described, the study concluded that higher urea concentrations are detrimental to the growth of *Arthrospira platensis* and, consequently, to phycocyanin production.

### 3.5. Optimization of Nitrogen Sources in Maximizing Response Variables

The main objective of this study was to determine the optimal parameters for urea and KNO_3_ concentration to maximize the growth of *Arthrospira platensis* biomass and phycocyanin production. In this assessment, the desirability function was used to establish a balance between the responses. The optimization process was carried out using the statistical software Design-Expert version 22.0.8, which defined desirability values from 0 to 1, where 0 represented the minimum responses and 1 the maximum responses for each model [40].

From the measurements of biomass and phycocyanin obtained from the 27 experimental runs during the experimental period (24 days), a general desirability value of 0.843 was obtained, indicating a high score in the response maximization process. The optimal concentration conditions for urea and KNO_3_ were −0.755 and 0.51 in coded units, respectively. These optimal conditions predict the following results for the response variables: biomass = 0.788 g L^−1^ and phycocyanin = 12.55%. Furthermore, the optimal values for KNO_3_ and urea for maximizing the response variables were 3.1 g L^−1^ and 0.098 g L^−1^, respectively. Table 4 shows the results generated in the optimization.

## 4. Conclusions

This experimental study was conducted to determine the optimal concentrations of nitrogen sources (urea and KNO_3_) to maximize the growth of *Arthrospira platensis* biomass and phycocyanin production. The results obtained through the response surface methodology were satisfactory for optimizing the independent variables and analyzing their effects on the response variables (biomass and phycocyanin). The reduced quadratic polynomial models demonstrated an adequate level of prediction accuracy. The optimal condition was predicted using the desirability function, resulting in a value of 0.843, indicating a suitable score in the optimization process. The primary objective of this study was to maximize *A. platensis* biomass growth and phycocyanin production. The optimized concentrations of urea and KNO_3_ using RSM were −0.755 and 0.51 in coded units, and the actual optimal values were 0.098 g/L and 3.1 g/L, respectively.

It is worth noting that urea concentrations were directly related to ammonia concentrations. It was observed that treatments with higher urea concentrations led to the death of microalgal biomass due to the toxicity of the culture medium. However, in treatments with low urea levels, biomass and phycocyanin production increased, which is attributed to the ability of *A. platensis* to assimilate this nitrogen source without high energy expenditure. During the study period, no significant nitrate consumption was recorded, as high ammonia concentrations caused by urea in the culture media inhibited the specific nitrate transporters (NRTs) inside the cyanobacteria. Finally, further studies are recommended to better understand the behavior of the factors studied, particularly urea. Additionally, it is advisable to evaluate lower urea levels to avoid medium toxicity and achieve better results in terms of biomass growth and phycocyanin production.

## Figures and Tables

**Figure 1 microorganisms-12-00363-f001:**
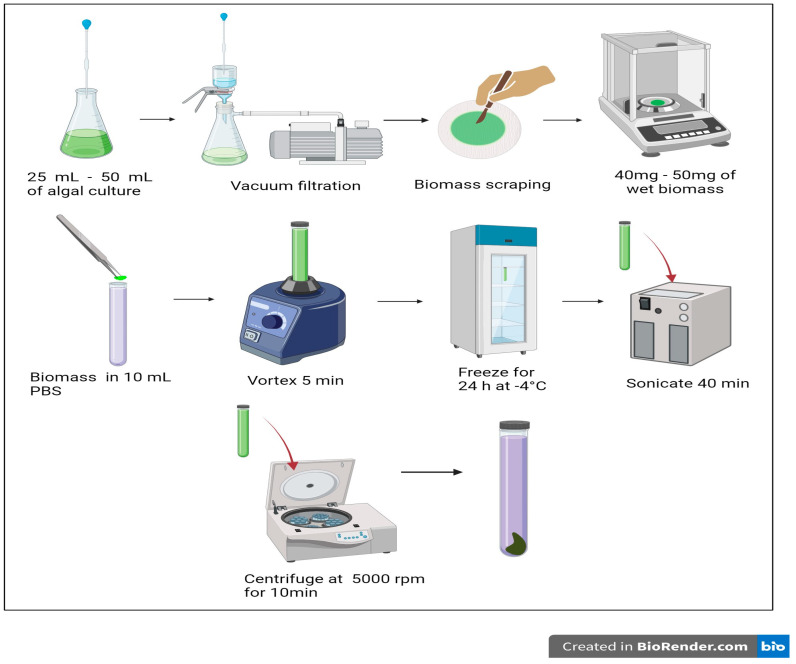
Phycocyanin extraction process from *Arthrospira platensis* wet biomass.

**Figure 2 microorganisms-12-00363-f002:**
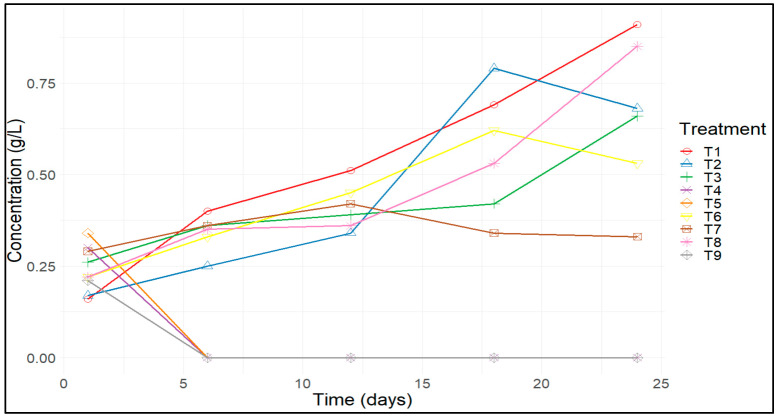
Growth curves of *Arthrospira platensis* over the 24 days of cultivation.

**Figure 3 microorganisms-12-00363-f003:**
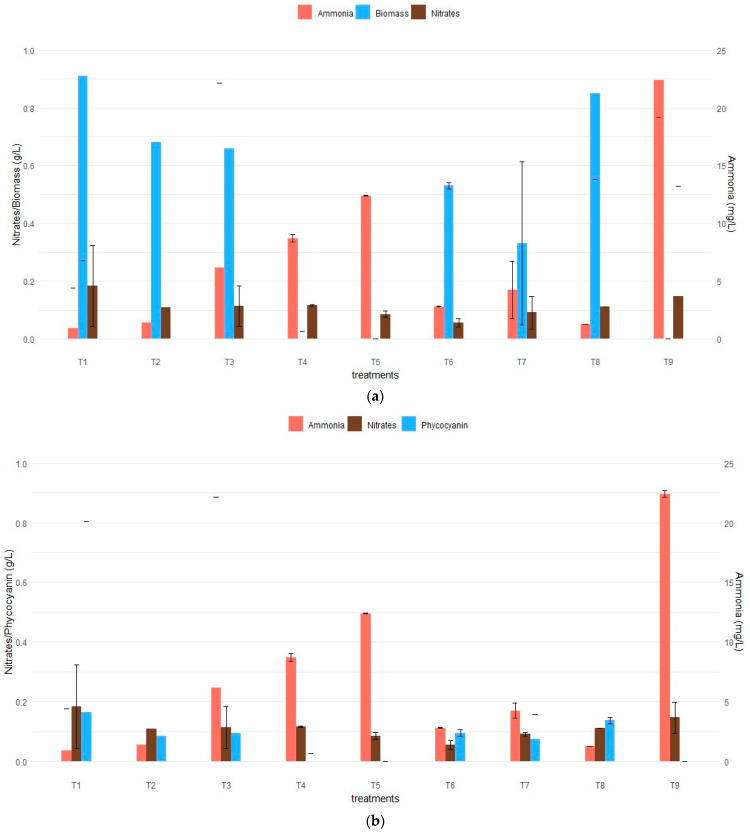
Relationship between nitrate and ammonia concentrations with respect to response variables: (**a**) biomass growth considering nitrate and ammonia concentrations in the culture medium; (**b**) phycocyanin production considering nitrate and ammonia concentrations in the different CCD treatments.

**Figure 4 microorganisms-12-00363-f004:**
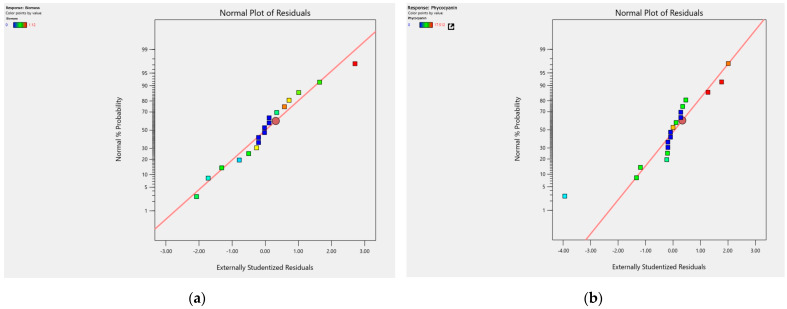
Normal probability plots for residuals: (**a**) plot for residuals of the reduced quadratic model for biomass; (**b**) plot for residuals of the reduced quadratic model for phycocyanin.

**Figure 5 microorganisms-12-00363-f005:**
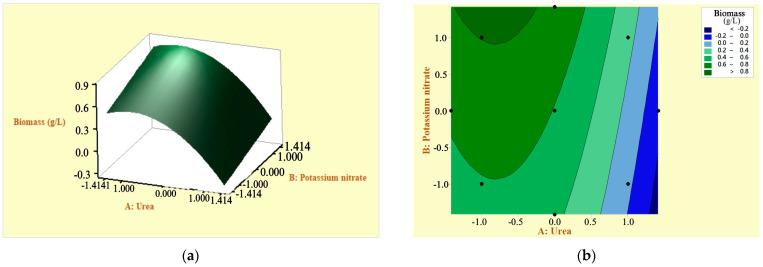
Response surface methodology based on CCD: (**a**) response surface plot of biomass considering the factors urea and KNO_3_; (**b**) contour plots of biomass considering the factors urea and KNO_3_; (**c**) response surface plot for phycocyanin; (**d**) response contour curves for phycocyanin production considering the factors urea and KNO_3_.

**Table 1 microorganisms-12-00363-t001:** Controllable factors and level coding.

Factors	Levels				
	−1.4142	−1	0	1	1.4142
KNO_3_ (g L^−1^)	0.5	1.5	2.5	3.5	4.5
Urea (g L^−1^)	0.01	0.08	0.15	0.23	0.31

**Table 2 microorganisms-12-00363-t002:** Central composite design (CCD) and results on day 24.

Treatments	Coded Units		Original Units		Biomass (g L^−1^)	Phycocyanin (%)
	KNO_3_	Urea	KNO_3_	Urea		
1	1.4142	0	4.5	0.15	0.91	15.87
2	0	0	2.5	0.15	0.68	8.68
3	1	−1	3.5	0.08	0.66	9.83
4	0	1.4142	2.5	0.31	0.00	0.00
5	−1	1	1.5	0.23	0.00	0.00
6	−1.4142	0	0.5	0.15	0.53	10.44
7	−1	−1	1.5	0.08	0.33	7.12
8	0	−1.4142	2.5	0.01	0.85	13.65
9	1	1	3.5	0.23	0.00	0.00

**Table 3 microorganisms-12-00363-t003:** ANOVA analysis of central composite design (CCD) models for optimizing KNO_3_ and urea concentrations for *Arthrospira platensis* growth and phycocyanin production.

Response	Model	F-Value	*p*-Value	*p*-Value Lack of Adjustment	R^2^	R^2^_adj_
Biomass	Reduced quadratic	11.34	0.0005	0.6211	0.7907	0.7210
Phycocyanin	Reduced quadratic	10.78	0.0006	0.7146	0.7822	0.7096

**Table 4 microorganisms-12-00363-t004:** Results from factor optimization and response prediction values.

Factors: Coded Values		Actual Optimal Values		Optimization of Results		General Desirability
Urea	KNO_3_	Urea (g L^−1^)	KNO_3_ (g L^−1^)	Biomass	Phycocyanin	
−0.755	0.51	3.1	0.098	0.6211	0.7907	0.843

## Data Availability

The data presented in this study are available on request from the corresponding author.

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
