# Peer review of "Optimization of an Alternative Culture Medium for Phycocyanin Production from Arthrospira platensis under Laboratory Conditions"

_microorganisms, 2024, doi:10.3390/microorganisms12020363_

Round 1
Reviewer 1 Report
Comments and Suggestions for Authors
This ms use RSM in a CCD to obtain the optimal composition of the culture medium for A. platensis. Authors concluded that the optimal concentrations of KNO3 and urea were 0.098g/L and 3.1g/L. The current results provide valuable information for improving the biomass and phycocyanin production of A. platensis.
Some specific comments are listed below
1 In the introduction, line 48-52, more information about the utilization of different nitrogen should be added.
2 In table 1, how does authors determine the concentration of KNO3 and urea used here?
3 Line 145, KNO3 should be KNO3, check this throughout the paper.
4 Line 166-167, urea used in treatment 1 is 0.15 gL-1, not 0.01 gL-1.
5 Line 180-186, most of mentioned paper about the inhibitory concentration of urea is over than 0.3 gL-1, however, 0.23 gL-1 used in the present study lead to death of A. platensis, why? Same question in line 216-220, 0.18 gL-1 was recognized as an inhibitory concentration. What’s the difference between these papers and which concentration should be the “correct” one?
6. The quality of all presented figures should be largely improved, including the lable size, error bars, resolution, etc.
7. In 3.2, it's better to provide more information about the equation 7 and 8.
Comments on the Quality of English Language
The ms is well written and no comment for the quality of English.
Author Response
Hi, please see the attachment.

Reviewer 2 Report
Comments and Suggestions for Authors
Reviewer comments
Manuscript: Optimization of an alternative culture medium for phycocyanin production from Arthrospira platensis under laboratory conditions
ID: microorganisms-2756855
Date: 2023-12-27
General comments
The author proposed an experimental investigation of the impact of several nitrogen source on biomass and phycosyanin production by Athropsira platensis. They used design of experiment approach to generate a response surface. While of merit, their investigations suffer from some lack of technical details and, more troubling a dramatic experimental error.
Major concerns
1. Line 35 – 38, please specify on which model the numerous benefits you cite were obtained. Indeed, in vitro findings do not have the same strength as those in animal models, or humans.
2. Table 1, the value of α is WRONG! From the value the authors used, it should be -2. Please redo all the calculation and graphs accordingly (or the experiments with an adequate 1.4142 level, but that would take too much time).
3. Line 91-95, was the inoculum pooled then split between the 27 runs? If yes, that is a good practice, please make it clearer. In addition, specify here that the runs were biologically triplicated.
4. In the same way, specify if all the runs were carried out in parallel? Within the same incubator? Specify the culture vessel.
5. Line 197-205, the authors mention ammonia. Yet, pH plays an important role in the balance between ammonium and ammonia. Was pH measured? If yes, please report the value.
6. Figure 5, the surfaces are not drawn over the entire domain (-α to +α). Please provide response surface spanning over the whole domain.
7. Explain how “general desirability” was computed. Indeed, from Figure 5, the maxima for both biomass and phycocynanin are located as -1 and +1. The authors provides far different results.
8. Finally, as measurements were triplicated, standard deviations should be reported and commented. Please do.
Minor concerns
Line 46, “some lipids” would be better than “lipids”
Line 76, please do not use Lux, but µmolPhotonPAR/m²/s instead.
Table 1, the value of α should be given
Typos
Line 145 and 245, 3 should be a subscript in KNO3
Author Response
Hi, please see the attachment.

Reviewer 3 Report
Comments and Suggestions for Authors
Please see the attached document.

Author Response
Hi, please see the attachment.

Round 2
Reviewer 2 Report
Comments and Suggestions for Authors
Dear Authors,
Thank you for taking the reply to all the points I raised.
Sincerely yours,
Reviewer 3 Report
Comments and Suggestions for Authors
The revised manuscript can be accepted to publication in Microorganisms.